# Plant-Derived Bone Substitute Presents Effective Osteointegration in Several Clinical Settings: A Pilot Study from a Single Center

**DOI:** 10.3390/bioengineering12080861

**Published:** 2025-08-11

**Authors:** Gianluca Conza, Adriano Braile, Antonio Davide Vittoria, Nicola Di Cristofaro, Annalisa Itro, Gabriele Martin, Gabriella Toro, Pier Francesco Indelli, Vincenzo Salini, Giuseppe Toro

**Affiliations:** 1Multidisciplinary Department of Medical and Surgical Specialties and Dentistry, University of Campania “Luigi Vanvitelli”, 80138 Naples, Italy; gianluca.conza@studenti.unicampania.it (G.C.); adriano.braile@hotmail.it (A.B.); antoniodavide.vittoria@unicampania.it (A.D.V.); nicola.dicristofaro1@studenti.unicampania.it (N.D.C.); annalisa.itro@unicampania.it (A.I.); gabriele.martin@unicampania.it (G.M.); 2Department of Clinical Sciences and Translational Medicine, University of Rome Tor Vergata, 00133 Rome, Italy; 3Unit of Radiology, San Paolo Hospital, 80125 Naples, Italy; gabriella.toro@tiscali.it; 4Department of Orthopaedic Surgery, Südtiroler Sanitätsbetrieb, 39042 Brixen, Italy; pindelli@stanford.edu; 5Department of Orthopaedic Surgery, Stanford University School of Medicine, Redwood City, CA 94061, USA; 6Facoltà di Medicina e Chirurgia, Università Vita-Salute San Raffaele, Via Olgettina 58, 20132 Milan, Italy

**Keywords:** bone graft, rattan wood, bone substitute, osteointegration, TCP, calcium phosphate, orthopedics

## Abstract

Background: Bone loss management is a tough challenge in orthopedic and trauma surgery that is generally treated using graft or substitute. Bone is the second most common transplanted tissue behind blood. Autologous bone graft represents the gold standard, while allograft is generally used as a secondary option, considering their impressive osteoconductive and osteoinductive properties. However, both allograft and autograft sources are limited. Therefore, synthetic bone substitutes gained popularity due to their low cost and ease of application. β-tri-Calcium phosphate (β-TCP) is a promising material implemented as a bone substitute. One of the limits of bone substitutes is related to their three-dimensional organization, which rarely replicates that of the normal bone. b.Bone™ is a novel bone substitute derived from rattan wood with a unique 3D structure that mimics the architecture of the human bone. This study aims to objectively evaluate the osteointegration of b.Bone™ in complex clinical settings. Methods: We retrospectively evaluated eight patients who underwent surgeries requiring filling bone loss through the use of b.Bone™. Osteointegration of the bone substitute was evaluated radiologically using a modified Van Hemert classification. Results: Eight patients were enrolled into this study: five females and three males with a mean age of 53,75 years old. b.Bone™ was applied in the following shapes: granules in four cases, cylinders in three cases and a prism in one. In four patients, the osteointegration reached a grade Van Hemert 4, three a grade 3, and only one a grade 2. Conclusions: β-TCP-based bone substitutes, such as those derived from rattan, appear to facilitate successful osteointegration in various clinical settings. Future studies with larger cohorts and longer follow-ups are necessary to evaluate the long-term efficacy of this promising substitute.

## 1. Introduction

Bone loss is a tough challenge in both orthopedics and trauma surgery that may complicate surgery and affect outcomes [1]. Bone grafts play a paramount role in these settings, representing about 2 million procedures worldwide, making it the second most transplanted tissue behind blood [2]. Autologous bone graft is still considered the gold standard for the treatment of bone loss thanks to its properties of osteoconduction, osteoinduction, and osteogenesis [3]. However, the limitation of autograft sources and donor-site morbidity are some of the factors reducing its routine use [4]. For these reasons, bone allografts are very often preferred [5]. However, their use is not risk-free, and their sources are limited. Therefore, bone substitutes are becoming a very common choice [4]. Several types of bone substitutes are currently available [4]. Calcium-based ceramics are among the most used bone substitutes. They are mainly composed of a mixture of Hydroxyapatite (HA) (in detail: Ca_10_(PO_4_)6(OH)_2_) and Tricalcium Phosphate (TCP) (in detail: Ca_3_(PO_4_)_2_)) [6]. β-TCP was originally developed for dentistry use but has reached a prominent role in orthopedic surgery over the last decades [7]. One of the limitations of bone substitutes is related to their three-dimensional organization, which rarely replicates that of normal bone. Considering the need for an adequate structural niche for reliable cellular function, the absence of a proper 3D organization may lead to poor osteointegration and outcomes [8]. Recently, a plant-derived bone substitute (b.Bone™ (GreenBone s.p.a., Faenza, Italy)) was developed to overcome these limitations. In particular, it is a HA- and β-TCP-based substitute derived from the biomorphic transformation of rattan wood, leading to a peculiar porous and highly interconnected 3D structure that carefully mimics human bone hierarchical architecture [9]. This bone substitute has been proven to be effective in in vitro studies and in simple clinical applications [9,10]. However, its properties make it feasible, especially in complex scenarios [11].

Therefore, we reported the radiographic results of b.Bone™ in complex and mixed cases performed at our institute to better understand its integration and stability.

## 2. Materials and Methods

We conducted a retrospective review of patients surgically treated in complex clinical situations involving bone gap filling. Inclusion criteria were: (1) age over 18; (2) bone defects of 5 cm or less; (3) no signs of infection; (4) proper soft tissue coverage, namely soft tissue injury between 0 to 2 as defined by the Nerve Ischemia Soft tissue Skeletal Shock Age (NISSSA) score [12]; and (5) application of b.Bone™ (GreenBone s.p.a., Faenza, Italy) regardless of available shapes.

Exclusion criteria included (1) age under 18 years old; (2) active infection according to Metsemakers et al. [13] for fracture-related infections and the International Consensus Meeting criteria for periprosthetic joint infection [14]; (3) bone defect larger than 5 cm; and (4) inadequate tissue coverage (namely NISSSA > 2) [12]. See Table 1 for further details. In all cases, b.Bone™ was augmented using bone-marrow-derived concentrated mesenchymal stem cells. b.Bone™ is sterilized through gamma ray irradiation, a process that appears not to affect cell replication [10,15]. BM-MSCs were harvested from the iliac crest and concentrated using the SEPAX 2 processing system as previously reported (Biosafe Group SA, Eysins, Switzerland) [16,17]. This system utilizes software-assisted physical principles, including Ficoll density gradient separation, to isolate and concentrate cellular components in a sterile, closed environment using disposable single-use kits [16]. It enables the automated, consistent, and operator-independent preparation of concentrated bone marrow aspirate [17].

Data on patient demographics, comorbidities, diagnosis, type of surgery, as well as the shapes of the bone substitutes and their dimensions were collected. The osteointegration and remodeling of the graft were radiographically evaluated by analyzing anterior-posterior and lateral X-rays performed at regular intervals. Three experts (P.I., V.S., and G.T.), two of whom were not involved in the surgical procedures (P.I. and V.S.), used a modified Van Hemert classification to assess b.Bone™ osteointegration [18] (see Table 2 for further details). When one of the three evaluators disagreed with the others, the bone osteointegration was classified based on consensus. To evaluate differences in the shape of b.Bone™, a statistical analysis was performed using one-way ANOVA, Welch’s *t*-test, and the Kruskal–Wallis H-test for non-parametric variables. Specifically, three different time points with the most available data were selected: 1 month, 2 months, and 10 months after surgery. One-way ANOVA tested for differences in measurement values across graft shape groups at each time point. An independent *t*-test was used to analyze differences between granules and cylinder shapes at the same time points. Spearman correlation test was used to evaluate correlation between follow-up time and Van Hemmert grading. Note that the prism group was not included in the analysis, as it consisted of only one patient. Significance was set at *p* < 0.005. This study complies with the Declaration of Helsinki, as revised in 2013. All patients provided written informed consent authorizing surgical management and data collection for research and audit purposes. Data included in this study were collected following our institutional ethical approval (ID #0020791/i). This article was written and developed in accordance with the Strengthening the Reporting of Observational Studies in Epidemiology (STROBE) checklist for cohort studies [19].

## 3. Results

Eight patients were included in this study, comprising three males and five females. The mean age was 53.75 years (range 19 to 84, SD ± 18.97). b.Bone™ was applied in different shapes: granules in four cases, cylindrical in three cases, and prismatic in one case. Patients’ diagnoses, types of surgery, b.Bone™ shapes, and dimensions are summarized in Table 3. The mean radiological follow-up was 9.8 months (range 4 to 23, SD ± 6.87). Radiological assessment of b.Bone™ osseointegration at the last follow-up showed grade 4 in 4 patients, grade 3 in 2 patients, and grade 2 in 2 patients. Examples of different shapes of b.Bone™ substitute applications are shown in Figure 1, Figure 2, Figure 3 and Figure 4. Their Van Hemert grade variations over time are summarized in Figure 5, Figure 6 and Figure 7.

The one-way ANOVA revealed that the type of bone graft shape has a significant influence on osteointegration. In fact, since the first month after the surgery, a significant difference was observed between the means of the different shapes (F(3, 17) = 4.53, *p* = 0.0165) This effect became more pronounced at 2 months (F(3, 12) = 15.75, *p* = 0.0004) and especially at 10 months (F(2, 10) = 45.08, *p* = 0.00004), suggesting that the choice of shape may significantly influence the osteointegration. These results are summarized in Table 4. The results of the statistical analysis conducted to better understand changes in osteointegration over time dependent on b.Bone™ shape are summarized in Table 5. For granules, both ANOVA (F = 21.54, *p* < 0.000001) and the Kruskal–Wallis test (H = 19.01, *p* = 0.00078) indicated highly significant variation in measurement values across timepoints. The Spearman correlation test confirmed a significant increase in Van Hemmert grading over the time of follow-up, with the granules demonstrating the strongest correlation (see Table 6 and Figure 8).

## 4. Discussion

Bone defect management is often complex, especially when associated with severe conditions like joint arthroplasty loosening, high-energy musculoskeletal trauma, and fracture-related infections [21]. β-TCP has been gaining increasing interest in orthopedic surgery from both industry and practitioners. Clinical reports on β-TCP are continually improving, and its applications have been documented across various clinical settings. In fact, Loveland et al. showed promising results when using β-TCP in ankle and hindfoot fusion [22]. Specifically, the authors conducted a retrospective evaluation of using a β-TCP substitute combined with recombinant human platelet-derived growth factor (rhPDGF-BB) [22]. This study suggested that this combination was effective in achieving fusion, even among patients with severe comorbidities and a high risk of nonunion [22]. Similar results were reported by DiGiovanni et al. in a prospective randomized controlled study involving 434 patients [23].

Although autografts and allografts remain the primary sources of bone grafts required in serious clinical situations like fracture nonunions [24], β-TCP-based bone substitutes, which can be ready to use even in complex conditions [25], are gaining popularity due to their properties very close to those hypothesized for an ideal bone substitute (i.e., good osteoconduction and osteoinduction capacities) [8,21]. Recently, the Diamond Concept, conceived by Giannoudis, has focused on graft selection and biological factors or activators for fracture healing [26]. β-TCP substitutes enhanced with biological elements like mesenchymal stem cells (MSCs) could be a game changer in managing bone defects, thanks to the neo-osteogenic and neo-angiogenic properties of MSCs [27,28,29]. Zheng et al. reported the effectiveness of β-TCP, added to bone marrow MSCs and macrophage cultures, in enhancing osteogenic differentiation of stem cells through macrophage polarization and *Wnt* pathway regulation [30]. Furthermore, a calcium phosphate bone substitute was retrospectively studied in a cohort of 290 patients treated for fracture fixation [31], showing a reduction in nonunion incidents [31]. Sasaki et al. reviewed 35 cases of lower limb reconstruction using the induced membrane technique (IMT) and bone gap filling with β-TCP [32], reporting successful bone healing in all cases regardless of defect size [32]. One of the limitations of current bone substitutes is related to their three-dimensional shape, which rarely replicates that of normal bone [33]. This may lead to poor outcomes because, to differentiate properly, a cell needs an appropriate three-dimensional niche [33]. More specifically, β-TCP scaffolds could induce bone remodeling by enhancing the expression of genes involved in osteoclast differentiation through extracellular space pathways [33].

To overcome this limitation, a plant-derived bone substitute (b.Bone™) has recently been developed. Another key characteristic of this bone substitute is porosity optimization, a characteristic that furtherly supports cell migration and binding into the scaffold [34]. In fact, a higher porosity has been shown to enhance cell proliferation by facilitating the transport of oxygen and nutrients [35]. In contrast, lower porosity has been associated with increased bone turnover, as evidenced by higher expressions of alkaline phosphatase and osteocalcin [35]. However, higher material porosity results in decreased mechanical strength [30,35]. Therefore, a balance between optimal porosity and mechanical strength must be achieved [30,35]. These features might be provided by b.Bone™ and, together with the three-dimensional structure, could be crucial for successful radiographical integration and satisfactory outcomes. The 3D structure of bone scaffolds creates a symbiosis with the organism [33]. In a recent paper by Ji et al., the transcriptome of three main types of 3D scaffolds—natural polymer hydrogel (gelatin-methacryloyl, GelMA), synthetic polymer (polycaprolactone, PCL), and bioceramic (β-TCP)—was analyzed [34]. The authors used RNA-seq analysis to examine the characteristics of the symbiosis niche developed during bone healing [33]. All three substitutes can promote bone healing, but β-TCP displayed a unique pattern of gene activation that could potentially accelerate the process. These findings were further confirmed by observing the effects of the β-TCP environment on osteoclasts: lysosome and antigen processing were upregulated, while extracellular matrix production and chondrocyte differentiation were downregulated, pathways closely linked to bone remodeling [33]. Some researchers suggest that hydrogels could also be enhanced in terms of gene activation when supplemented with bone marrow MSCs [36]. Several studies have focused on the behavior of 3D-printed scaffolds concerning cell proliferation and differentiation. The general consensus is that 3D scaffolds can have cellular permeability similar to natural bone and higher mechanical properties [37]. Therefore, a bone substitute that mimics the in vivo complexity of bone tissue in terms of physico-chemical, topological, and structural properties could induce the formation and remodeling of mechanically competent bone tissue [38]. Reproducing these characteristics with 3D-printed scaffolds is challenging, and production costs are significant [38]. Another approach to improving 3D scaffolds involves organoids, which are multicellular clusters derived from stem or progenitor cells [39]. Recent research suggests that combining bioprinting with organoid cultures could better replicate the complex interactions between cells and the microenvironment, promising for regenerative medicine [39,40]. The concept of three-dimensionality has recently advanced in materials engineering [41]. An evolving technology in this field is represented by 4D implants [41], which upgrade 3D implants by incorporating “smart” materials with shape memory properties, making them dynamic [41]. Li et al. reported their experience with this innovative technology in treating a cavitary bone defect of the distal femur caused by osteoblastoma, achieving good radiographical and clinical outcomes at 12 months [41]. This technology could be a significant step forward in filling irregularly shaped bone defects [41]. However, like all new technologies, considerations regarding costs and availability must be addressed before widespread adoption. To overcome these challenges, some researchers are exploring the processing of natural materials with structural features similar to bone rather than fabricating artificial ones [42]. One method involves the biomorphic transformation of natural wood structures into 3D porous calcium phosphate ceramics. Rattan wood has been used as a template for this process because of its hierarchical structure, resembling the osteon structure of long bones [43]. Alt et al. reported a series of nine patients who underwent bone defect filling with b.Bone™ following iliac crest tricortical bone graft harvesting, with no complications such as postoperative hematoma, surgical revision, or implant-related issues [9]. The authors observed good radiographical integration after an average follow-up of 9.8 months [9]. Another notable application of b.Bone™ was described by Pitsilos and Giannoudis in a case involving a cylinder augmented with bone marrow concentrate to repair a 2.2 cm bone defect resulting from atrophic nonunion after an open fracture of the distal femur, with good outcomes [44]. However, to the best of our knowledge, none of the studies in the available literature have adequately described radiological osteointegration using a graded and validated assessment tool. In our study, all patients showed good osteointegration with b.Bone™, regardless of the shape used. Nonetheless, granules may undergo more substantial changes over time, potentially making them more susceptible to biological remodeling that influences radiologic evaluation. Conversely, for the cylinder group, ANOVA revealed an infinite F-statistic (F = infinity, *p* < 0.0001) due to zero variance within the timepoints. Despite this, the Kruskal–Wallis test still detected a statistically significant difference (H = 19.00, *p* = 0.00416). These combined findings suggest that even seemingly stable materials can show measurable progression over time. Overall, these results indicate that both the graft shape and performance change over time, with granules exhibiting more time-dependent variability and cylinders appearing more stable or possibly plateauing. However, as represented by the Spearman correlation test, the small sample size underpowered the results of the cylinder group, limiting the precision of the observed correlation.

Our study has some limitations, mainly related to the small patient cohort and the absence of a control group that underpowered the statistical analysis. However, the combination of parametric and non-parametric tests may enhance the robustness of our conclusions despite data limitations such as small or uneven group sizes. Additionally, to our knowledge, a specific radiological grading system for β-TCP grafts in scenarios other than knee osteotomies is not available. Therefore, we had to adapt the Van Hemert grading system for our purposes. Another limitation of this study was that we did not evaluate the resorption kinetics of b.Bone™, although this was outside the main objectives. Finally, the innovative nature of the bone substitute and its potential use in complex scenarios prompted us to report these preliminary results. Further studies with larger cohorts are necessary to validate our findings and, hopefully, lead to the development of a radiological grading system specifically designed for β-TCP scaffold integration.

## 5. Conclusions

The present study supports b.Bone™ as a useful bone substitute with reliable osteointegration properties also in complex clinical scenarios. Our findings suggest that b.Bone™ is suitable for different surgical treatments and facilitates radiological bone healing without adverse reactions, making it a promising alternative to autografts and other synthetic substitutes. However, studies on resorption kinetics and with larger cohorts and longer follow-up are needed to validate its efficacy.

## Figures and Tables

**Figure 1 bioengineering-12-00861-f001:**
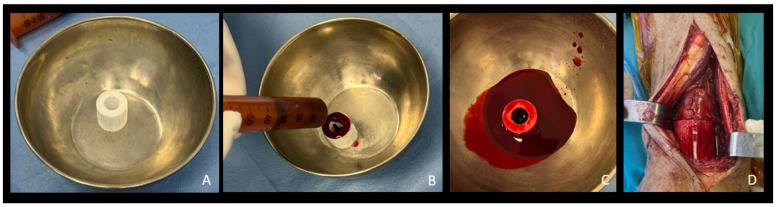
Intraoperative pictures of a case of distal tibia septic nonunion treated with b.Bone™ cylinder application to fill the bone gap (patient F.S.) (**A**) b.Bone™ cylinder; (**B**,**C**) addition of BM-MSCs; (**D**) application of the cylinder to fill the bone gap.

**Figure 2 bioengineering-12-00861-f002:**
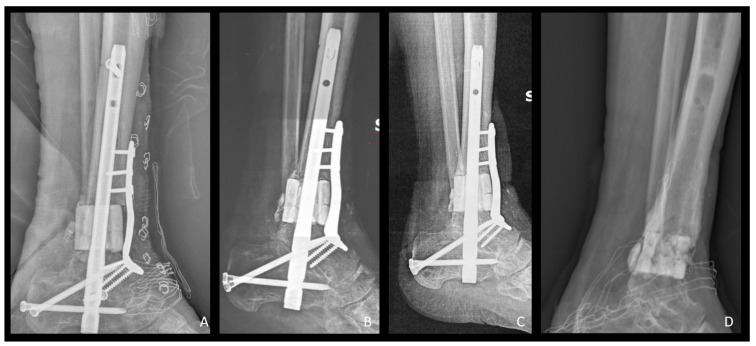
Lateral view X-Rays representing radiological follow-up of the same patient as in Figure 1. (**A**) Postoperative X-Ray showing b.Bone™ cylinder application. Fixation was achieved through a retrograde ankle fusion nail and an anterior plate; (**B**–**D**) follow-up at 6, 12, and 23 months after the surgery. Note: in (**D**), b.Bone™ integration and stability even after hardware removal.

**Figure 3 bioengineering-12-00861-f003:**
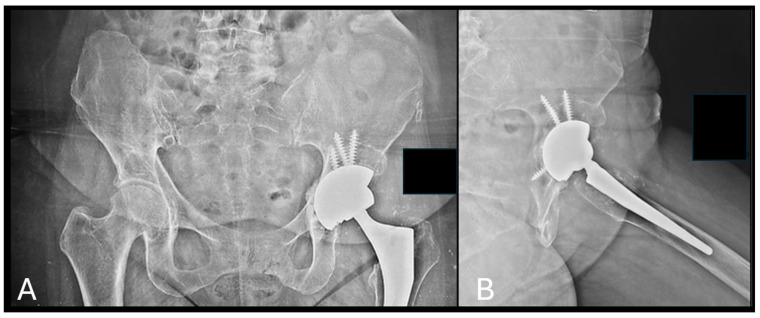
Application of b.Bone™ granules in a case of acetabular revision for aseptic loosening at 6 months after the surgery. (**A**) Pelvis AP view; (**B**) left hip axial view.

**Figure 4 bioengineering-12-00861-f004:**
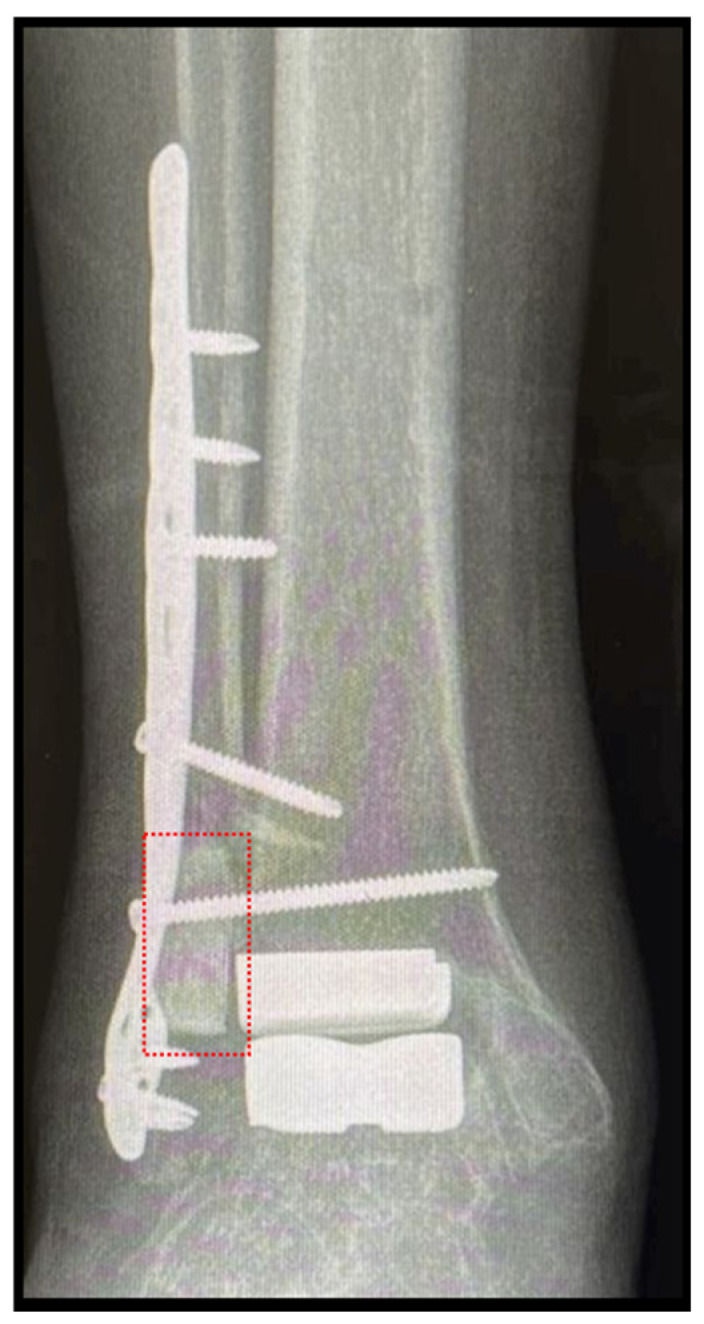
Total ankle replacement through lateral approach in a case of ankle ankylosis. Note: application of the b.Bone™ prism (box in red) to fill the bone gap occurred during the transfibular approach. Follow-up at 4 months.

**Figure 5 bioengineering-12-00861-f005:**
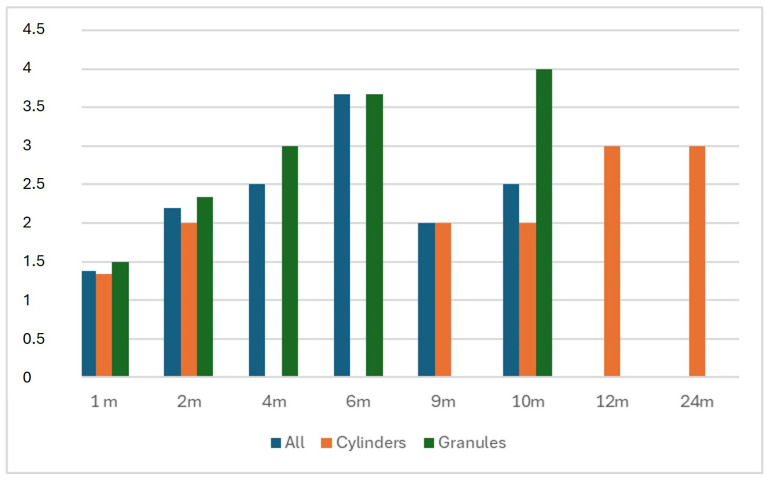
Variation of Van Hemert grades [18] attributed by the three observers through radiographic evaluation at regular timepoints. On the horizontal axis: time (months); on the vertical axis: Van Hemert grades.

**Figure 6 bioengineering-12-00861-f006:**
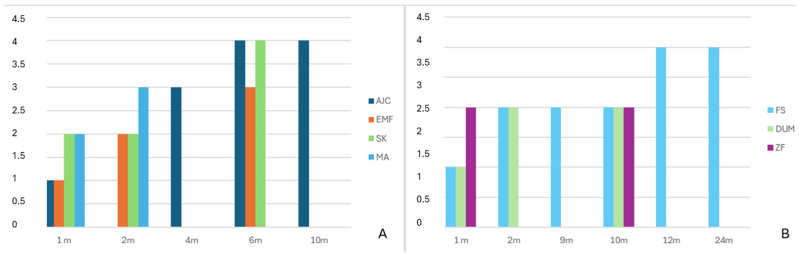
Van Hemert classification [18] throughout the entire follow-up period according to single patients. (**A**) Granules; (**B**) cylinders.

**Figure 7 bioengineering-12-00861-f007:**
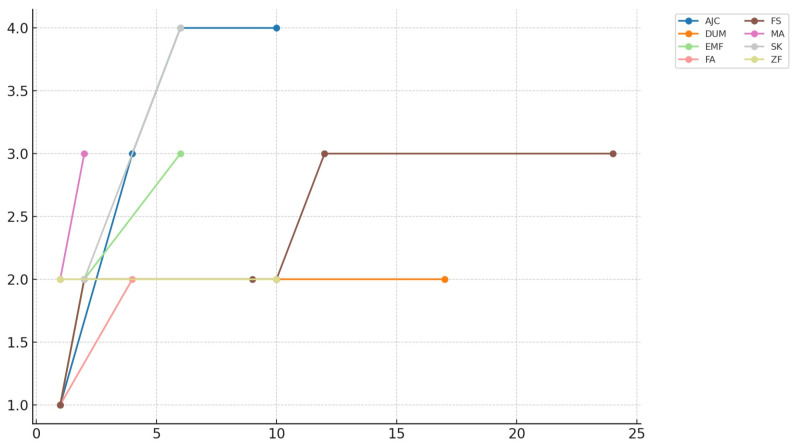
Patient-specific Van Hemert grading variations through the entire follow-up. On the horizontal axis: time (months); on the vertical axis: Van Hemert grades.

**Figure 8 bioengineering-12-00861-f008:**
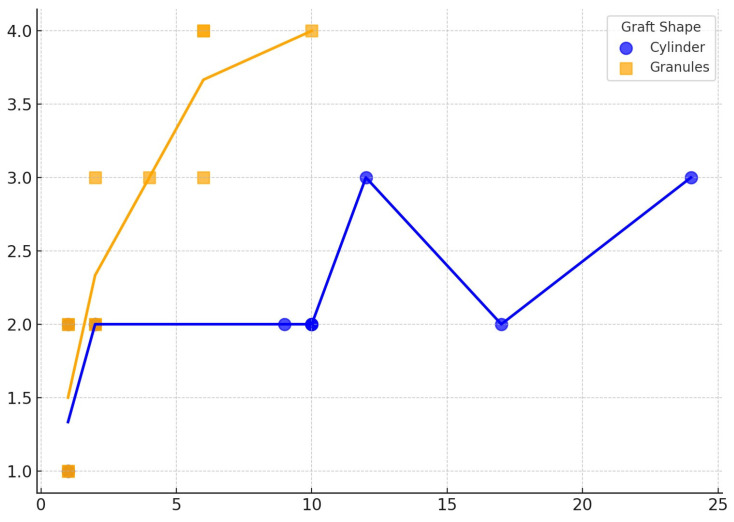
Graphical representation of the Spearman correlation test representing the degree of osteointegration (namely Van Hemmert grading) throughout the entire follow-up and the shape of bone substitute used. Both graft shapes demonstrated progressive osteointegration with time, with the strongest association observed in the granule cohort. On the horizontal axis: time (months); on the vertical axis: Van Hemert grades.

**Table 1 bioengineering-12-00861-t001:** Indications for b.Bone™ use in the included population.

- Bone defects with a maximum size of 5 cm
- No signs of infection (according to Metsemakers et al. [13] for fracture-related infections and the International Consensus Meeting criteria for periprosthetic joint infection [14])
- Proper soft tissue coverage (soft tissue injury between 0 and 2 as defined by the Nerve Ischemia Soft tissue Skeletal Shock Age (NISSSA) score [12])
- Age Over 18 years

**Table 2 bioengineering-12-00861-t002:** Modified Van Hemert classification as proposed by Ozalay et al. [18] of biphasic calcium phosphate (BCP) graft remodeling used for knee osteotomy.

Phases of BCP Remodeling	Radiological Definition
0—Direct postoperative	No sign
1—Vascular phase	Rounded osteotomy sites, clear distinction between BCP and bone
2—Osteoblastic phase	Whitened osteotomy sites, distinction between BCP and bone still visible
3—Consolidation phase	Distinction between BCP and bone not visible, cloudy bone formation
4A—Full reformation	No sign of osteotomy, BCP visible
4B—Full reformation	Disappearance of BCP

**Table 3 bioengineering-12-00861-t003:** General characteristics of the analyzed population, including surgical procedures, the graft used, and the Van Hemert grading at final follow-up. **N.a.:** not available. **TKA:** Total knee arthroplasty. **THA:** Total hip arthroplasty. **ORIF:** Open reduction and internal fixation. **TAA:** Total ankle arthroplasty.

Patient	Sex	Age	Comorbidities	Diagnosis	Surgical Treatment	Graft Shape (*Dimension in cm*)	Follow-Up (Months)	Van Hemert Grading
M.A.	F	66	Rheumatoid arthritis, type II diabetes, obesity	Aseptic loosening of TKA + periprosthetic fracture	Revision TKA + ORIF	Granules (*n.a.*)	4	3
A.J.C.	F	52	n.a.	Pelvic instability	Pubic fusion	Granules (*n.a.*)	10	4
E.M.F.	M	63	Smoker, chronic liver disease, previous drug abuse	Septic arthritis	Two-stage THA	Granules (*n.a.*)	5	3
D.U.M.	M	84	Former smoker, hypertension	Distal tibia osteomyelitis	Resection and ankle fusion	Cylinder (*5*)	17	2
S.K.	F	56	n.a.	Aseptic loosening of THA	Acetabular revision	Granules (*n.a.*)	6	4
F.S.	M	45	Smoker, previous drug abuse	Distal tibia osteomyelitis	Resection and ankle fusion	Cylinder (*3*)	23	3
Z.F.	M	19	Neurofibromatosis type 1	Skewfoot	Medial column lengthening	Cylinder (*1.7*)	10	2
F.A.	M	45	Previous drug abuse, rheumatoid arthritis, HIV infection, chronic liver disease	Ankle ankylosis	TAA	Prism (*2.4*)	4	2

**Table 4 bioengineering-12-00861-t004:** One-way ANOVA and Welch’s *t*-test used for the evaluation of time-dependent changes in Van Hemert classification [18] of the b.Bone™. ANOVA F: One-Way ANOVA with F distribution [20].

Timepoint	ANOVA F (df)	ANOVA *p*-Value	Welch’s *t*-Test t	Welch’s *t*-Test *p*-Value
1 m	4.53 (3.17)	0.0165	2.65	0.0331
2 m	15.75 (3.12)	0.0004	1.58	0.1747
10 m	45.08 (2.10)	4.4 × 10^−5^	∞	<0.0001

**Table 5 bioengineering-12-00861-t005:** Results of the one-way ANOVA and Kruskal–Wallis test to evaluate time-dependent changes between cylinders and granules. ANOVA F: One-Way ANOVA with F distribution [20].

Shape	ANOVA F (df)	ANOVA *p*-Value	Kruskal–Wallis H	Kruskal–Wallis *p*-Value
Granules	21.54 (4.8)	>0.000001	19.01	0.00078
Cylinder	∞ (3, 5)	>0.0001	19.00	0.00416

**Table 6 bioengineering-12-00861-t006:** Spearman correlation between follow-up time and Van Hemert grading for osteointegration, stratified by graft shape. Both cylinders and granules demonstrated significant positive correlations, indicating progressive improvement in osteointegration over time. Confidence intervals were derived using 1000 bootstrap resamples. **Note:** NaN indicates that the confidence interval could not be estimated due to limited sample size. Although the correlation coefficient and *p*-value remain valid, the precision is limited.

Graft Shape	Spearman ρ	*p*-Value	95% CI
Cylinder	0.765	0.0038	nan–nan
Granules	0.905	0.0001	0.772–0.972

## Data Availability

Data available on request due to restrictions (The included patients are part of a more large study that we are conducting at our institute, in which are included other patients and also biochemical markers. The full data of the included patients are protected by the study design and the full data are managed by other researchers of our group.).

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
