# Peer review of "Plant-Derived Bone Substitute Presents Effective Osteointegration in Several Clinical Settings: A Pilot Study from a Single Center"

_bioengineering, 2025, doi:10.3390/bioengineering12080861_

Round 1
Reviewer 1 Report
Comments and Suggestions for Authors
This study addresses a clinically relevant and timely topic by investigating the osteointegration potential of a novel, plant-derived β-TCP bone substitute (b.Bone) in complex orthopedic scenarios. The material's biomorphic structure and derivation from rattan wood offer an innovative approach to overcoming the limitations of conventional bone grafts, particularly regarding structural mimicry and biocompatibility. But it needs to Major Revisions before publication.
1. The current manuscript lacks sufficient experimental depth and logical coherence. Several key sections, particularly the methodology and discussion, are underdeveloped, and narrative transitions are abrupt, making the argumentation less persuasive.
2. The small sample size and absence of a control group significantly limit the generalizability of the findings, and there is a need for more comprehensive statistical analysis to substantiate the observed trends.
3. The methodology and results require greater detail—for instance, in describing radiographic evaluation procedures, graft application protocols—to fully support the study's conclusions.
4. While the modified Van Hemert grading system was applied to assess osteointegration, its use outside of knee osteotomy cases is not fully justified or validated, have questions about the scoring’s reproducibility.
5. To strengthen the discussion and provide a broader scientific context, the authors are encouraged to cite additional recent literature, such as, doi:10.12336/bmt.25.00009, 10.36922/OR025040004, 10.12336/biomatertransl.2024.03.003, 10.36922/OR025040005.
Author Response
- The current manuscript lacks sufficient experimental depth and logical coherence. Several key sections, particularly the methodology and discussion, are underdeveloped, and narrative transitions are abrupt, making the argumentation less persuasive.
R: We developed the discussion and the methodoloy as you suggested.
2 The small sample size and absence of a control group significantly limit the generalizability of the findings, and there is a need for more comprehensive statistical analysis to substantiate the observed trends.
We performed the statistical analysis as you seggested.
3. The methodology and results require greater detail—for instance, in describing radiographic evaluation procedures, graft application protocols—to fully support the study's conclusions.
We revised as you suggested
4. While the modified Van Hemert grading system was applied to assess osteointegration, its use outside of knee osteotomy cases is not fully justified or validated, have questions about the scoring’s reproducibility.
We aknowledge that the nature of the Van Hemert Grading is not specifically conceived for this kind of evaluation, as we have written in the manuscript. But, to the best of our knowledge, is the most close and suitable for this work. Anyway this limitation has been already added in our limitations section.
5. To strengthen the discussion and provide a broader scientific context, the authors are encouraged to cite additional recent literature, such as, doi:10.12336/bmt.25.00009, 10.36922/OR025040004, 10.12336/biomatertransl.2024.03.003, 10.36922/OR025040005.
We revised as you suggested.
Reviewer 2 Report
Comments and Suggestions for Authors
Ln 39. The opening statement, “Bone loss is a severe complication in both orthopedics and trauma surgery…” can be interpretted to suggest “orthopedics and trauma surgery” cause, or is responsible for, “bone loss”! I am sure this is not the authors intension; however, this is a common grammatical error for authors from non-native English-speaking backgrounds. Please consider: how might I captivate the reader? Why should the reader read any further? This statement also fails to differentiate critical-size and non-critical-size bone loss. The reader’s question is: ‘so what’?
Ln 46. The authors self-contradict: “bone allografts are very often preferred…(because) autograft source is limited and donor-site morbidity…reduc(e) its routinely use” (sic). These are not the reasons “bone allografts are very often preferred, accounting for nearly a third of all bone grafts performed in North America!” (sic) Of note, the North American orthopedic market is not the worlds largest! The reader asks again: ‘so what’?
Ln 51. Would the average reader comprehend; what is the difference between “hydroxyapatite (HA) and Tricalcium Phosphate (TCP)”? i.e. Ca3(PO4)2 (TCP) and Ca10(PO4)6(OH)2 (HA).
Ln 57. What is the evidence “b.Bone is a HA and β-TCP based substitute derived from the biomorphic transformation of rattan wood… that carefully mimics human’s bone hierarchical architecture”? (sic) A reference is required.
FYI, “b.Bone” is a Trademarked product; requires identification with ™.
Ln 61 The claim that “This bone substitute has already proved to be effective in in vitro studies and in simple clinical applications” (sic) is not substantiated by the cited work ( )The cited work reports surgically created “defects at the iliac crest” (i.e. non-load-bearing) in 9 patients treated with “Greenbone scaffold biomaterial (i.e. b.Bone™)… press-fitted clear into the defect” (sic). In vitrostudies are not reported.
Ln 75. For the purposes of reproducibility and verification by others, please clinical and laboratory details for “bone-marrow derived mesenchymal stem cells concentrated. This latter was harvested from the iliac crest and concentrated using the Sepax cell-separation system.” (sic)
Lns 119, 120, 124, 128, 135, 138, 145, 151. In each line, mention is made of β-TCP. However, β-TCP is a heterogenous, structurally diverse, product. With each mention it is not clear which specific β-TCP is being discussed! This is not reproducible.
It is nice to read correctly formatted References! Thank you.

It is clear the authors native language is not English.
There are several instances of incorrect grammar, creating ambiguity and miscomprehension. It will require sub-editorial intervention.
Author Response
1: Ln 39. The opening statement, “Bone loss is a severe complication in both orthopedics and trauma surgery…” can be interpretted to suggest “orthopedics and trauma surgery” cause, or is responsible for, “bone loss”! I am sure this is not the authors intension; however, this is a common grammatical error for authors from non-native English-speaking backgrounds. Please consider: how might I captivate the reader? Why should the reader read any further? This statement also fails to differentiate critical-size and non-critical-size bone loss. The reader’s question is: ‘so what’?
R: we revised the introduction section
2. Ln 46. The authors self-contradict: “bone allografts are very often preferred…(because) autograft source is limited and donor-site morbidity…reduc(e) its routinely use” (sic). These are not the reasons “bone allografts are very often preferred, accounting for nearly a third of all bone grafts performed in North America!” (sic) Of note, the North American orthopedic market is not the worlds largest! The reader asks again: ‘so what’?
R. We revised as you suggested.
3. Would the average reader comprehend; what is the difference between “hydroxyapatite (HA) and Tricalcium Phosphate (TCP)”? i.e. Ca3(PO4)2 (TCP) and Ca10(PO4)6(OH)2 (HA).
R: thank you for your suggestion, we revised as you suggested.
4: Ln 57. What is the evidence “b.Bone is a HA and β-TCP based substitute derived from the biomorphic transformation of rattan wood… that carefully mimics human’s bone hierarchical architecture”? (sic) A reference is required.
R: We revised as you suggested.
5:FYI, “b.Bone” is a Trademarked product; requires identification with ™.
<R: Thank you we revised as you suggested
6: Ln 61 The claim that “This bone substitute has already proved to be effective in in vitro studies and in simple clinical applications” (sic) is not substantiated by the cited work ( )The cited work reports surgically created “defects at the iliac crest” (i.e. non-load-bearing) in 9 patients treated with “Greenbone scaffold biomaterial (i.e. b.Bone™)… press-fitted clear into the defect” (sic). In vitrostudies are not reported.
R: we revised as you suggested.
7: Ln 75. For the purposes of reproducibility and verification by others, please clinical and laboratory details for “bone-marrow derived mesenchymal stem cells concentrated. This latter was harvested from the iliac crest and concentrated using the Sepax cell-separation system.” (sic)
R: we improved the laboratory details as you suggested
8: Lns 119, 120, 124, 128, 135, 138, 145, 151. In each line, mention is made of β-TCP. However, β-TCP is a heterogenous, structurally diverse, product. With each mention it is not clear which specific β-TCP is being discussed! This is not reproducible.
R: we agree with you but not in every cited study the type of BTCP used has been specified. When available we added more details.
Reviewer 3 Report
Comments and Suggestions for Authors
The manuscript presents a promising plant-derived bone substitute, b.Bone, demonstrating successful osteointegration in clinical settings. However, the small sample size and lack of a control group limit the generalizability of the findings. The adaptation of the Van Hemert grading system to evaluate osteointegration may not fully reflect the unique characteristics of b.Bone, and a specific radiological grading system for β-TCP substitutes would strengthen the study. Larger cohorts and long-term follow-ups are essential to validate the long-term efficacy and consistency of b.Bone.
- The authors provide detailed inclusion/exclusion criteria regarding patient comorbidities (e.g., diabetes, osteoporosis, smoking) which may affect healing?
- What was the average time from surgery to radiological evidence of osteointegration for each patient? Please provide a timeline.
- Was there any correlation between graft shape (granules, cylinders, prism) and osteointegration outcomes? Please provide statistical analysis.
- Only descriptive statistics are presented. Can the authors include inferential statistics to determine significance between graft shapes and Van Hemert scores?
- Authors include confidence intervals and standard deviations for the osteointegration scores across patients?
- Was any degradation analysis of b.Bone performed during the follow-up period to assess resorption kinetics?
- What sterilization process was used for b.Bone, and could this affect its bioactivity or MSC adhesion?
- clarify the specific indications (trauma, infection, arthroplasty failure) and whether outcomes differed between these subgroups.
- the data in Table 3 be presented visually (e.g., heatmap, bar chart) to better reflect trends in integration and follow-up time?
Author Response
1:
- The authors provide detailed inclusion/exclusion criteria regarding patient comorbidities (e.g., diabetes, osteoporosis, smoking) which may affect healing?
R: We added details about comorbidities, thank you for your suggestion
2. What was the average time from surgery to radiological evidence of osteointegration for each patient? Please provide a timeline.
We illustrated the evolution of osteointegration trhough time in fig. 4 and 5
3. Was there any correlation between graft shape (granules, cylinders, prism) and osteointegration outcomes? Please provide statistical analysis.
R. we provided the statistical analysis as you suggested.
4.Only descriptive statistics are presented. Can the authors include inferential statistics to determine significance between graft shapes and Van Hemert scores?
R. we improved the statistical analysis as you suggested.
5: Authors include confidence intervals and standard deviations for the osteointegration scores across patients?
When available we improved this section
6. Was any degradation analysis of b.Bone performed during the follow-up period to assess resorption kinetics?
Unfortunately, We do not have any data about resorption kinetics. we added this in the limitations section.
7. The product is sterilized through gamma rays. In vitro studies reported that the b.Bone is biocompatible and has no negative effect on cells biological activity (REF Alt, V.; Walter, N.; Rupp, M.; Begué, T.; Plecko, M. Bone Defect Filling with a Novel Rattan-Wood Based Not-Sintered Hydroxyapatite and Beta-Tricalcium Phosphate Material (b.BoneTM) after Tricortical Bone Graft Harvesting – A Consecutive Clinical Case Series of 9 Patients. Trauma Case Rep 2023, 44, 100805, doi:10.1016/j.tcr.2023.100805.
Minardi, S.; Corradetti, B.; Taraballi, F.; Sandri, M.; Van Eps, J.; Cabrera, F.J.; Weiner, B.K.; Tampieri, A.; Tasciotti, E. Evaluation of the Osteoinductive Potential of a Bio-Inspired Scaffold Mimicking the Osteogenic Niche for Bone Augmentation. Biomaterials 2015, 62, 128–137, doi:10.1016/j.biomaterials.2015.05.011)
8. clarify the specific indications (trauma, infection, arthroplasty failure) and whether outcomes differed between these subgroups.
R: DIagnosis are already reported in table 3
9. the data in Table 3 be presented visually (e.g., heatmap, bar chart) to better reflect trends in integration and follow-up time?
Thank you for your suggestion we have included a diagram about the osteointegration evaluation throughout the followup.
Round 2
Reviewer 1 Report
Comments and Suggestions for Authors thanks for your replyAuthor Response
Thank you.
Reviewer 2 Report
Comments and Suggestions for Authors
This is a revision of work I reviewed previously, and recommended ‘major revision’, primarily because inadequate detailed materials and methods did not facilitate reproducibility by others. While some of these details have now been included, these are generalized; they do not detail exactly how the authors acquired their reported results. The existence of many editing widows illustrates the lack of attention to detail and disrespect for the reader.
Ln 17. Punctuation. The sentence: “Bone loss management is a tough challenge in orthopaedic and trauma surgery. ,generally treated using graft or substitute.” (sic) is poor grammar; editing widows (incorrect punctuation) are evident.
Ln 125. Figure 1 is a composite image, created from 6 individual images, and presented in two planes. Unfortunately, the figure legend fails to identify each (image) data source, instructing the reader “From left to right…” (sic). Two images are annotated “SN”; however, this is not identified in the figure legend. Scale/size bars are absent. Figure 1 does not meet basic requirements for reporting scientific data.
Ln 129. Figure 2 is a composite image, created from 2 individual images, presented without identify each data source. Are these the same patient? What is the orientation? What is the meaning of annotations? Scale/size bars are absent. How were these data acquired?
Ln 132. Figure 3 is a radiographic image of Total Ankle Replacement for ankylosis, illustrating the application of b.Bone™ prism applied to the fibula. However, additional features are evident in these data; the tallus (and navicular?) has also been replaced, and fibula fixed to the tibia. How does the uninformed reader identify the “b.Bone™ prism”?
Ln 133. Spelling. Should “ankylosisb.Bone™ prism” (sic) be ‘ankylosis. b.Bone™prism’?
Ln 135. Figure 4, “Van Hemert classification variation across all available timepoints” (sic) is uninterpretable. Axes are not labelled. Units/measures are not defined. To what data (i.e. source) was the Van Hemert classification applied?
Ln 151. Table 4 is uninterpretable. Please identify the source of data to which “one-way ANOVA and t-test (were) performed to assess differences across shapes throughout the timepoints” (sic)?
What is “ANOVA F (df)” (sic)? What is “t-test t” (sic)? What is the variable “shapes”?
Ln 213. Spelling. Should “MSCs[35].Several…” be ‘’MSCs [35]. Several…’?
Ln 253. Spelling. Should “influencesradiologicall” (sic) be ‘influences radiological’?
Ln 258. Spelling. Should “toghetr” (sic) be ‘together’?
Ln 262. The statement “the combination of parametric and non-parametric tests may improve the robustness of these conclusions…” illustrates misunderstanding of the fundamentals of scientific interpretation. Robustness is a property of data; it is not a property of the “conclusions”! Data is “parametric” (i.e. data distribution is known and is based on a fixed set of parameters – often ‘a normal distribution’), or it is “non-parametric” (i.e. data distribution is unknown, and parameters are not fixed.) The application of “parametric and non-parametric tests” (sic) is dictated by properties of the data.
Ln 269. Spelling. Should “…study.Finally…” (sic) be ‘…study. Finally…’
Ln 274. This reader does not understand how “reliable osteointegration properties” ensure “an effective bone substitute” (sic)? Surely an effective bone substitute must exhibit more than radiological integration? Bone substitutes must also exhibit functional substitution of bone tissue?

Overlooking the high incidence of editing widows (spelling and punctuation), the language is understandable.
Author Response
- Ln 17. Punctuation. The sentence: “Bone loss management is a tough challenge in orthopaedic and trauma surgery. ,generally treated using graft or substitute.” (sic) is poor grammar; editing widows (incorrect punctuation) are evident.
R: we revised as you suggested.
- Ln 125. Figure 1 is a composite image, created from 6 individual images, and presented in two planes. Unfortunately, the figure legend fails to identify each (image) data source, instructing the reader “From left to right…” (sic). Two images are annotated “SN”; however, this is not identified in the figure legend. Scale/size bars are absent. Figure 1 does not meet basic requirements for reporting scientific data.
R: Dear reviewer, We improved the figure and legend as you suggested.
- Ln 129. Figure 2 is a composite image, created from 2 individual images, presented without identify each data source. Are these the same patient? What is the orientation? What is the meaning of annotations? Scale/size bars are absent. How were these data acquired?
R: Dear reviewer, We improved the figure and legend as you suggested.
- Ln 132. Figure 3 is a radiographic image of Total Ankle Replacement for ankylosis, illustrating the application of b.Bone™ prism applied to the fibula. However, additional features are evident in these data; the tallus (and navicular?) has also been replaced, and fibula fixed to the tibia. How does the uninformed reader identify the “b.Bone™ prism”?
R: Dear reviewer, We improved the figure and legend as you suggested.
- Ln 133. Spelling. Should “ankylosisb.Bone™ prism” (sic) be ‘ankylosis. b.Bone™prism’?
R: We revised as you suggested thank you.
- Ln 135. Figure 4, “Van Hemert classification variation across all available timepoints” (sic) is uninterpretable. Axes are not labelled. Units/measures are not defined. To what data (i.e. source) was the Van Hemert classification applied?
R: We improved the legend as you suggested. Thank you.
- Ln 151. Table 4 is uninterpretable. Please identify the source of data to which “one-way ANOVA and t-test (were) performed to assess differences across shapes throughout the timepoints” (sic)?
We improved the legend to better understand the table but the data that were object of the analysis are described in the materials section. Thank you for your suggestion.
- What is “ANOVA F (df)” (sic)? What is “t-test t” (sic)? What is the variable “shapes”?
Dear reviewer thank for your question. the available shapes of the b.Bone have been discussed in the introduction and materials section, we believe that the reader can recall this information since there are also figures that show granules, prysm and cylinders. The F-statistic in ANOVA (Analysis of Variance) is a ratio: of between-group variance to within-group variance. We included the F-statistic, degrees of freedom and p-value to better describe the result of the ANOVA analysis. We also used independent samples t-test to furtherly deepen our analysis. Anyway we improved the legend in order to clarify the tables.
- Ln 213. Spelling. Should “MSCs[35].Several…” be ‘’MSCs [35]. Several…’?
R: We improved the legend as you suggested. Thank you.
- Ln 253. Spelling. Should “influencesradiologicall” (sic) be ‘influences radiological’?
R: We improved the legend as you suggested. Thank you.
- Ln 258. Spelling. Should “toghetr” (sic) be ‘together’?
R: We improved the legend as you suggested. Thank you.
- Ln 262. The statement “the combination of parametric and non-parametric tests may improve the robustness of these conclusions…” illustrates misunderstanding of the fundamentals of scientific interpretation. Robustness is a property of data; it is not a property of the “conclusions”! Data is “parametric” (i.e. data distribution is known and is based on a fixed set of parameters – often ‘a normal distribution’), or it is “non-parametric” (i.e. data distribution is unknown, and parameters are not fixed.) The application of “parametric and non-parametric tests” (sic) is dictated by properties of the data.
R: Dear reviewer thank you for your suggestion. We revised the sentence as you suggested.
We applied both parametric (ANOVA, t-test) and non-parametric (Kruskal-Wallis) methods to improve the strenght and reliability of our findings, given the characteristics of our dataset. Specifically:
-
Parametric tests (One-Way ANOVA and Welch’s t-test) were used to detect differences in group means, where assumptions of normality and variance homogeneity were reasonably met or adjusted for (e.g., using Welch’s t-test for unequal variances).
-
Non-parametric tests (Kruskal-Wallis) were included to validate the results, particularly in contexts where:
-
Sample sizes were small,
-
Data were unbalanced or skewed,
-
Some groups had zero variance (e.g., cylinder group with identical values at certain timepoints),
making parametric assumptions more fragile.
-
- Ln 269. Spelling. Should “…study.Finally…” (sic) be ‘…study. Finally…’
R: We revised as you suggested, thank you.
- Ln 274. This reader does not understand how “reliable osteointegration properties” ensure “an effective bone substitute” (sic)? Surely an effective bone substitute must exhibit more than radiological integration? Bone substitutes must also exhibit functional substitution of bone tissue?
Dear reviewer we revised as you suggested.
Reviewer 3 Report
Comments and Suggestions for Authors
Accept in present form
Author Response
Thank you.
Round 3
Reviewer 2 Report
Comments and Suggestions for Authors
Ln 75. What is the definition of “proper tissue coverage”? Please identify the source of “b.Bone™”
Ln 82. Please cite the source for “Metsemakers et al.” Please cite the source for “International Consensus Meeting”? (sic)
Ln 83. What is the definition of “inadequate tissue coverage”?
Ln 86. In order to facilitate replication by others, the method “BM-MSCs were harvested from the iliac crest and concentrated using the SEPAX 2 processing system” (sic) is incomplete. What was the method applied to “harvest… BM-MSC… from the iliac crest”? (sic) What was the method applied to aspirate BM-MSC (i.e. diluent? separation? storage?)
Ln 157. This reader was unable to locate “Figures (7) to 43” (sic) and “Figures 54 and 65”. (sic) This reader could only locate Figures 1-6! Please correct.
Ln 170. Ln 204. Ln 209. Ln 223. Please assist the reader and include the source for “modified Van Hemert grading”? Figures and Tables should be interpretable without needing to back refer to the main text; i.e. interpretable in isolation!
Ln 212. This reader is not familiar with the term “the firts month”. (sic)
Ln 237. Please cite the source for “Loveland et al.”? (sic)
Ln 242. Please cite the source for “DiGiovanni et al.”? (sic)
Ln 256. What is the evidence “β-TCP-based bone substitutes, which can be ready to use even in complex conditions, are gaining popularity due to their properties very close to those hypothesized for an ideal bone substitute (i.e., good osteoconduction and osteoinduction capacities)”? (sic)
Ln 259. Please cite the source for the “Diamond Concept, conceived by Giannoudis.”? (sic)
Ln 263. Please cite the source for “Zheng et al.”? (sic)
Ln 269. Please cite the source for “Sasaki et al.”? (sic)
Ln 298. Grammar. Non-sequitur. “To overcome this limitation, a plant-derived bone substitute has recently been developed. Porosity is another key characteristic in scaffold optimization, supporting cell migration into the scaffold and increasing the surface area available for cell binding” (sic) are unrelated concepts! Is this an editing widow?
Ln 298 – Ln 362. Grammar. This paragraph is unnecessarily lengthy. Multiple subjects / concepts are discussed, that would be easier for the reader to comprehend is discussed using a single paragraph for each subject / concept! (i.e. One subject / concept per paragraph!)
Ln 309. Please cite the source for “Ji et al.”? (sic)
Ln 333. Please cite the source for “Li et al.”? (sic)
Ln 343. Please cite the source for “Alt et al.”? (sic)
Ln 347. Please cite the source for “Pitsilos and Giannoudis”? (sic)
Ln 525. This reader was unable to locate the source document: “Scheda Tecnica TS B-Bone Rev. 0124 (1).Pdf”(sic)

Document requires sub-editing for grammar.
Author Response
Ln 75. What is the definition of “proper tissue coverage”? Please identify the source of “b.Bone™”
R: We revised as you suggested thank you.
Ln 83. What is the definition of “inadequate tissue coverage”?
R: We revised as you suggested thank you
Ln 86. In order to facilitate replication by others, the method “BM-MSCs were harvested from the iliac crest and concentrated using the SEPAX 2 processing system” (sic) is incomplete. What was the method applied to “harvest… BM-MSC… from the iliac crest”? (sic) What was the method applied to aspirate BM-MSC (i.e. diluent? separation? storage?)
R: The sepax process is furtherly described in the reference.
Ln 157. This reader was unable to locate “Figures (7) to 43” (sic) and “Figures 54 and 65”. (sic) This reader could only locate Figures 1-6! Please correct.
R: We revised as you suggested thank you.
Ln 170. Ln 204. Ln 209. Ln 223. Please assist the reader and include the source for “modified Van Hemert grading”? Figures and Tables should be interpretable without needing to back refer to the main text; i.e. interpretable in isolation!
R: we revised as you suggested thank you
Ln 212. This reader is not familiar with the term “the firts month”. (sic)
R: we revised as you suggested, thank you
Ln 237. Please cite the source for “Loveland et al.”? (sic)
R: we revised as you suggested, thank you
Ln 242. Please cite the source for “DiGiovanni et al.”? (sic)
R: we revised as you suggested, thank you
Ln 256. What is the evidence “β-TCP-based bone substitutes, which can be ready to use even in complex conditions, are gaining popularity due to their properties very close to those hypothesized for an ideal bone substitute (i.e., good osteoconduction and osteoinduction capacities)”? (sic)
R: we revised as you suggested, thank you
Ln 259. Please cite the source for the “Diamond Concept, conceived by Giannoudis.”? (sic)
R: we revised as you suggested, thank you
Ln 263. Please cite the source for “Zheng et al.”? (sic)
R: we revised as you suggested, thank you
Ln 269. Please cite the source for “Sasaki et al.”? (sic)
R: we revised as you suggested, thank you
Ln 298. Grammar. Non-sequitur. “To overcome this limitation, a plant-derived bone substitute has recently been developed. Porosity is another key characteristic in scaffold optimization, supporting cell migration into the scaffold and increasing the surface area available for cell binding” (sic) are unrelated concepts! Is this an editing widow?
R. R: we revised as you suggested, thank you
Ln 298 – Ln 362. Grammar. This paragraph is unnecessarily lengthy. Multiple subjects / concepts are discussed, that would be easier for the reader to comprehend is discussed using a single paragraph for each subject / concept! (i.e. One subject / concept per paragraph!)
R: dear reviewer thank you for your suggestion, we agree with you. However the addiction of these parts of the discussion were required from other reviewers.
Ln 309. Please cite the source for “Ji et al.”? (sic)
R: we revised as you suggested, thank you
Ln 333. Please cite the source for “Li et al.”? (sic)
R: we revised as you suggested, thank you
Ln 343. Please cite the source for “Alt et al.”? (sic)
R: we revised as you suggested, thank you
Ln 347. Please cite the source for “Pitsilos and Giannoudis”? (sic)
R: we revised as you suggested, thank you
Ln 525. This reader was unable to locate the source document: “Scheda Tecnica TS B-Bone Rev. 0124 (1).Pdf”(sic)
R: we revised as you suggested, thank you